# Simulating Power Generation from Photovoltaics in the Polish Power System Based on Ground Meteorological Measurements—First Tests Based on Transmission System Operator Data

**Jakub Jurasz [1,2,*]**, **Marcin Wdowikowski [3]** and **Mariusz Figurski [3,4]**

1   Department of Engineering Management, Faculty of Management, AGH University, 30-059 Cracow, Poland
2   School of Business, Society and Engineering, MDH University, 722-20 Västerås, Sweden
3   Institute of Meteorology and Water Management-National Research Institute, 01-673 Warsaw, Poland;
    marcin.wdowikowski@imgw.pl (M.W.); mariusz.figurski@pg.edu.pl (M.F.)
4   Faculty of Civil and Environmental Engineering, Gdansk University of Technology, 80-233 Gdansk, Poland
*   Correspondence: jurasz@agh.edu.pl

**Abstract:** The Polish power system is undergoing a slow process of transformation from coal to one that is renewables dominated. Although coal will remain a fundamental fuel in the coming years, the recent upsurge in installed capacity of photovoltaic (PV) systems should draw significant attention. Owning to the fact that the Polish Transmission System Operator recently published the PV hourly generation time series in this article, we aim to explore how well those can be modeled based on the meteorological measurements provided by the Institute of Meteorology and Water Management. The hourly time series of PV generation on a country level and irradiation, wind speed, and temperature measurements from 23 meteorological stations covering one month are used as inputs to create an artificial neural network. The analysis indicates that available measurements combined with artificial neural networks can simulate PV generation on a national level with a mean percentage error of 3.2%.

**Keywords:** photovoltaics; artificial neural networks; national power system

## 1. Introduction

The transformation of the power system is a continuous process, and to fully realize this we will need years of ongoing commitment and well-thought decisions on country and regional levels. In the past two years in the Polish power system, we could observe a significant increase in the installed capacity in both residential and commercial photovoltaic (PV) systems. To a set of interesting investments in PV capacity one could potentially include a 600 kWp system located near Porąbka-Żar pumped-storage hydropower station, 2.5 MW installation for waterworks in Szczecin, and 739 kW for 35 buildings belonging to a housing cooperative in Wrocław. The growing interest in PV systems can be linked to (a) increasing electricity prices, (b) the decreasing cost of PV systems, and (c) growing awareness of the impact of the energy generation sector on the natural environment. (Impact of PV systems from the perspective of Life Cycle Assessment should not be neglected. Readers are referred to other works strictly dedicated to this topic.)

The Polish Transmission System Operator (PSE) has been publishing wind generation data on an aggregated level for quite some time. These time series with an hourly time step are of great importance to visualize and analyze the variability [1] of wind generation with different time horizons. They can also be used to create forecasting or simulation models [2–5]. Recently, the PSE has also made



publicly available aggregated generation time series from photovoltaic installations located in Poland. These data create new opportunities for further research, including an analysis of the complementarity between renewable energy sources in Poland [6] or the use of available ground measurements to simulate PV generation on a country level. The second research direction can be later used to simulate and predict how the growing installed capacity in PV systems in Poland will affect power system operations. Based on available measurement data, and knowing the transmission system constraints, decisions can be made regarding the optimal distribution of renewable generators in the power system.

Considering the information above, the objective of this study is to conduct first tests with regard to the possibility of using ground measurements from meteorological stations to simulate the power generation from PV systems on a country level. Such research results can be found in the international literature both for PV and wind generation, using different kinds of inputs and simulation tools. For example, for Sweden reanalysis based on MERRA (Modern Era Retrospective-Analysis for Research and Applications), data sets have been used to effectively model a national fleet of wind turbines' power output [7]. Recently, Olauson [8] found that ERA5 data sets performed much better than the MERRA reanalysis sets mentioned previously [7]. Similar to the analysis presented in this work, Black et al. [9] used meteorological data and regression techniques to simulate a fleet of PV systems.

## 2. Materials and Methods

To simulate the energy generation from PV systems on a country level, the method based on an artificial neural network (ANN) was applied. ANNs are a computational system loosely inspired by biological neural networks. They belong to a group of artificial intelligence information processing paradigms that has gained immense popularity in recent years. Simulation or forecasting models based on ANNs have been successfully applied in various areas of cognitive and application research, including water-demand forecasting [10], lake water-level forecasting [11], renewables integration studies [12], in the area of color image identification and reconstruction [13], multi-core optic fibers [14], wind speed prediction [15], or, most importantly from this paper's perspective, in the areas of direct and global radiation prediction [16] and PV energy yield forecasting [17].

A model of feedforward neural network (NN) has been developed in Matlab 2019a software. The Levenberg–Marquardt method was used to optimize the weights and bias values [18]. The input data were divided into training, validation, and testing subsets in proportions of 70, 15, and 15. Since the length of the available time series was limited to one month, the data were divided so that the first 70% of hours was used to teach the neural network, followed by 15% to validate/supervise the teaching process and the remaining 15% to test the NN performance. The number of neurons in the hidden layer was selected following a brute-force approach, namely NNs with the number of hidden neurons k ranging from 1 to $n$, where $n$ is the number of input neurons being tested. In the literature, various approaches to solving this problem can be found [18,19]. However, considering the low computing effort to create an ANN, a brute-force approach in this particular case seems to be a justified choice.

As inputs, the PV generation time series covering the month of May (available at https://www.pse.pl/) and time series of wind speed, temperature, and irradiation for 23 meteorological stations located in Poland, obtained from the Institute of Meteorology and Water Management—National Research Institute (IMWM-NRI), were used. The locations of meteorological stations along with the equipment used are presented in Figure 1 and in Table 1. The data have an hourly time step. The nighttime hours and hours when the energy generation from the PV system was less than 5 MW were removed from the input data set. Such low values were removed because hourly temporal resolution does not take into account the spatial distribution of PV systems in Poland, and low generation periods occur during sunset and sunrise. The final data set consisted of 473 hourly records. The data were normalized to a range of 0–1 before the ANN creation procedure. The performance of ANN has been assessed based

on two commonly applied metrics, namely (MAPE) mean absolute percentage error (Equation (1)) and (RMSE) root-mean-square error (Equation (2)).

$$MAPE = \frac{1}{n}\sum_{i=1}^{n}\left|\frac{o_i - s_i}{o_i}\right| \tag{1}$$

$$RMSE = \left[\sum_{i=1}^{n}(o_i - s_i)^2/n\right]^{1/2} \tag{2}$$

where $n$—sample size, $o$—observed value, and $s$—simulated value.

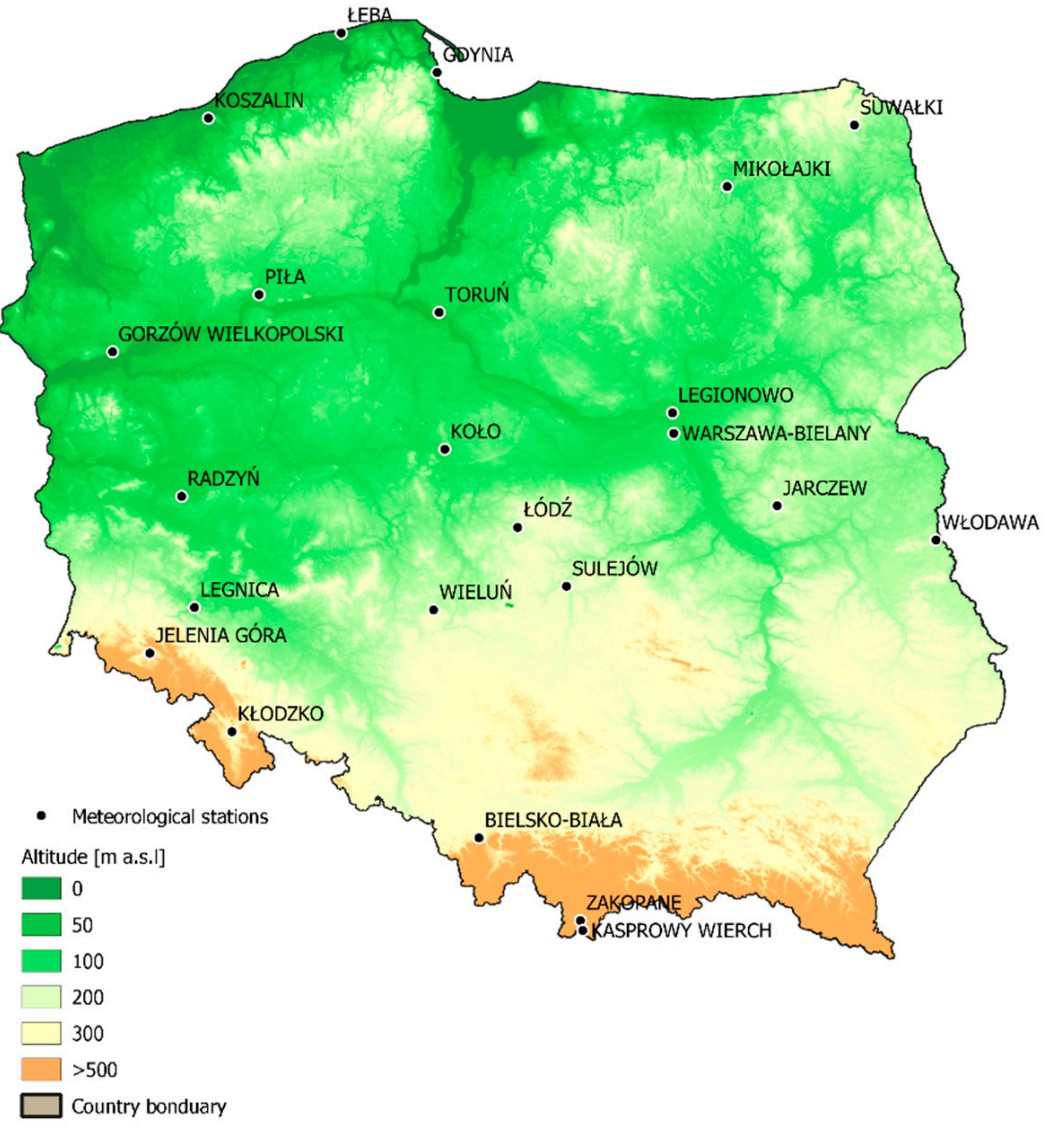

**Figure 1.** Locations of 23 IMWM-NRI meteorological stations.

**Table 1.** Meteorological stations measurement sensor description.

| No. | Meteorological Element | Sensor Type | | | |
|-----|------------------------|-------------|---|---|---|
| 1 | Irradiation | KIPP & ZONEN CNR4 Net Radiometer | KIPP & ZONEN CMP11 Pyranometer | X | X |
| | Location | Piła, Łeba | All 21 others | X | X |
| 2 | Air temperature | Pt 100 PVC | Vaisala QMT 103 | Vaisala QMT 110 | Vaisala HMP45A/45D |
| | Location | Radzyń, Warszawa-B., Legionowo, Jarczew, Kasprowy W., Wieluń, Zakopane, Łódź, Włodawa, Suwałki | Łeba, Koło, Bielsko-B., Piła, Gorzów W., Sulejów, Mikołajki | Toruń, Koszalin, Legnica, Jelenia G., Kłodzko | Gdynia |
| 3 | Wind speed | Vaisala Ultrasonic WS 425 | Vaisala Cup Anemometer WAA 151 | Gill Instruments WindSonicM | Others (Vaisala WMS 302, WMT 700, 702, 703 and G. Lufft WS 200-UMB) |
| | Location | Gorzów W., Jelenia G., Kłodzko, Koszalin, Legnica Łódź, Mikołajki, Piła, Sulejów, Suwałki, Wieluń, Włodawa, Zakopane | Gdynia, Radzyń, Warszawa-B. | Toruń, Legionowo | Jarczew, Kasprowy, Łeba, Koło, and Bielsko-B. |

## 3. Results and Discussion

The energy yield from photovoltaic modules is a function of irradiation falling on the module area, the modules' efficiency, and their temperature. Since detailed information about the PV systems' location is currently not available to the authors, nor is the specification of the modules used, it is justified to use a black-box approach where the input data are transferred into desired output. Because the temperature of the PV modules is determined by irradiation and ambient air temperature as well as wind speed, which has a cooling effect, the above-mentioned meteorological parameters have been considered as explanatory variables. To simulate an hourly power generation from PV systems in Poland, a set of 69 (3 meteorological parameters from 23 stations: Bielsko-Biała, Gdynia, Gorzów Wielkopolski, Jarczew, Jelenia Góra, Kasprowy Wierch, Kłodzko, Koło, Koszalin, Legionowo, Legnica, Łeba, Łódź, Mikołajki, Piła, Radzyń, Sulejów, Suwałki, Toruń, Warszawa-Bielany, Wieluń, Włodawa, and Zakopane – please see Figure 1) explanatory variables was used. These explanatory variables exhibited various correlation coefficient values with the response variable. For the irradiation, it was on average 0.777, whereas the air temperature and wind speed were, respectively, 0.439 and 0.299. Figure 2 shows the observed hourly irradiation on the 1st of May 2020 and the production of energy from PV systems at the national level. During that day, the observed irradiation in individual hours varied significantly among considered locations (meteorological stations), while the PV systems maintained a relatively smooth energy generation pattern. This phenomenon can be attributed to the spatial smoothing of power generation due to the geographical dispersion of the PV systems [20]. This situation is beneficial from the perspective of variable renewable energy sources (VRES) integration to the power system, although constraints such as transmission network capacity may limit the benefits resulting from spatial smoothing.

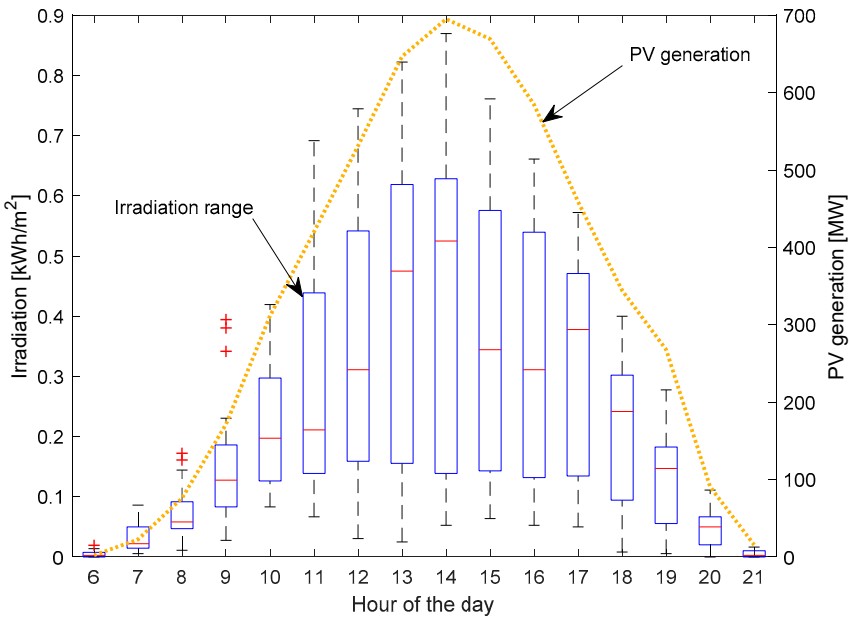

**Figure 2.** Observed irradiation in 23 meteorological stations along with photovoltaic (PV) production on 1st of May 2020.

The generation from PV systems during the whole period is visualized in Figure 3. Significant variability between individual daily yield sums can be observed. On a side note, accordingly to the PSE data in May, the PV systems generated 223 GWh, whereas wind turbines generated almost 1.072 TWh, which contributed to covering, respectively, 1.8% and 8.6% of the national demand in this month.

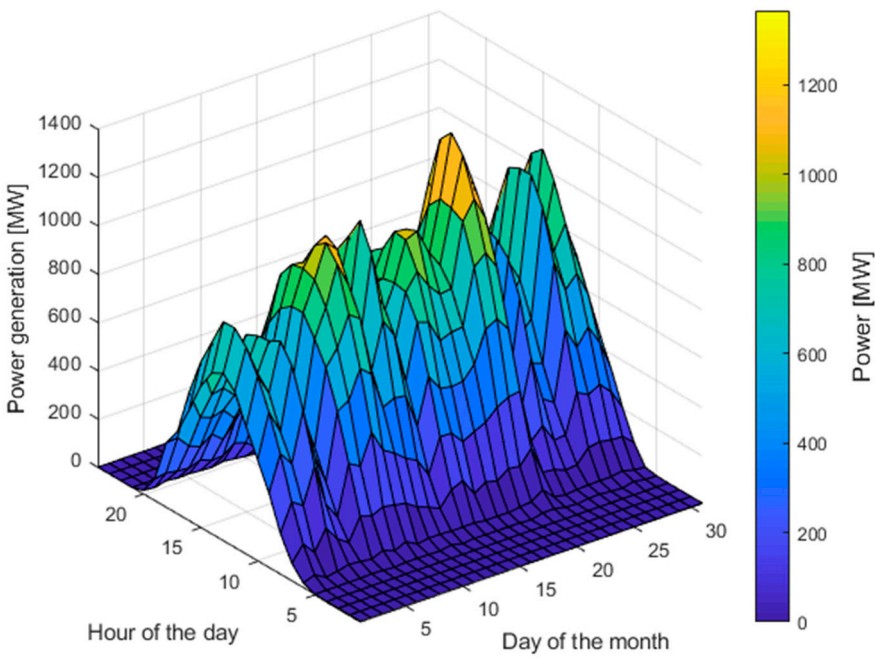

**Figure 3.** PV generation on a country level during May 2020.

As mentioned in the Methods and Data section, in total 69 potential configurations of the ANNs were tested. For those, the one with the lowest MAPE was selected for further analysis. The performance of the selected and the remaining ANNs as a function of the number of neurons in the hidden layer is presented in Figure 4. The best performing ANN had 15 neurons in the hidden layer and MAPE of 3.2%.

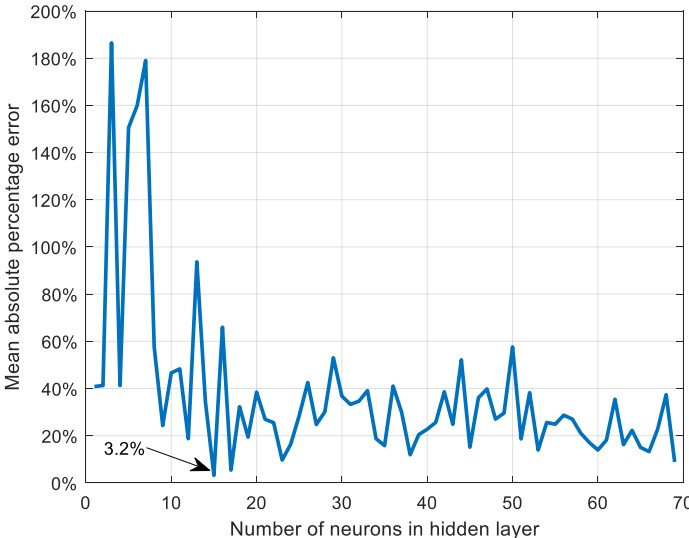

**Figure 4.** Performance of neural networks for the testing subset based on mean absolute percentage error (MAPE) criteria.

The ANN also performed well in terms of RMSE, which was found to be 39.2 MW. In Figure 5, the performance of the ANN was visualized for the testing subset only. This set is the final verification if the neural network is capable of obtaining good-quality results for input data that remained unknown during the training phase. In Figure 5 it can be noted that the ANN performed very well for the extreme values (very low and high generation), whereas some systematic errors with an unknown source occurred for the mid-range values. On average, it was found that the ANN tended to overestimate the generation by 6 MW. The highest overestimation error was found to be 133 MW, whereas the highest underestimation was 150 MW.

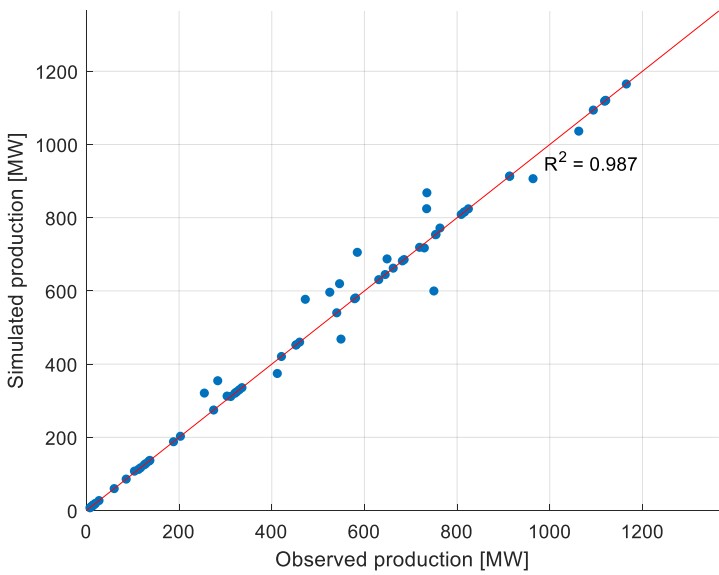

**Figure 5.** Performance of the neural network for the testing subset. $R^2$ refers to linear regression. Red line indicates a theoretical perfect match.

Figure 6 visualizes the performance of the neural network over a period of 4.5 days by the end of May 2020. The night hours are excluded from the analysis. As shown in the figure, the values simulated by the ANN followed the real PV systems generation well. During the second day, the ANN wrongly simulated a sudden drop in PV generation, increasing the variability of the modeled time

series (in terms of ramp rates). During the fourth day, one can observe that in the midday hours the simulated generation was slightly greater than the observed one. In general, the absolute errors did not exceed 150 MW.

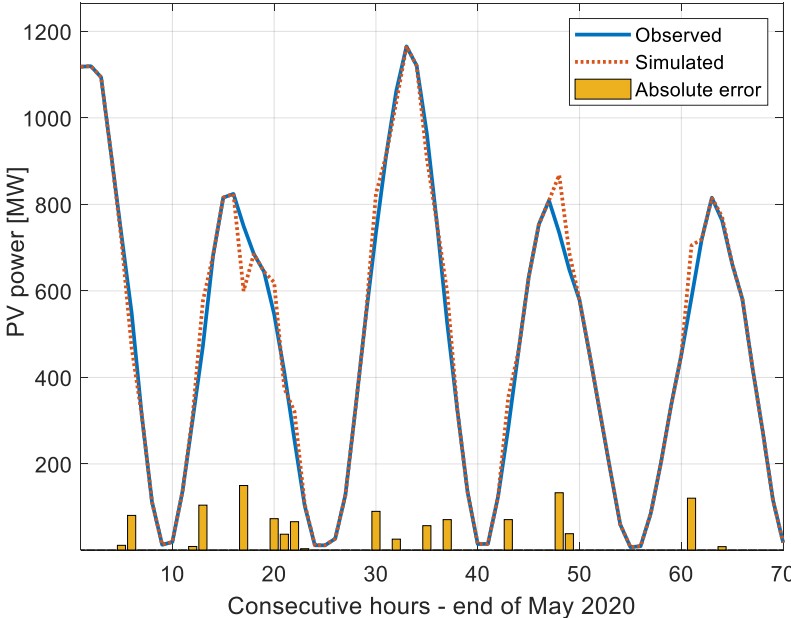

**Figure 6.** Performance of the neural network for the testing subset. Please note that night hours (no irradiation) are excluded.

## 4. Conclusions

The presented short communication archives the first results regarding the potential of simulating the generation from the PV systems on a national level based on ground measurements provided by the Institute of Meteorology and Water Management. The conducted analysis based on artificial neural networks revealed that ground measurements consisting of irradiation, wind speed, and air temperature can be used to correctly model the power generation from PV systems on a country level. Despite some limitations, such as neglecting the technical specification of the PV systems, the spatial distribution of PV farms across the country, taking input irradiation on a horizontal rather than an inclined surface, and finally representing the meteorological conditions in Poland based on a sample of 23 locations, it was possible to model the PV generation with a mean absolute percentage error of roughly 3.2%.

Most importantly, the obtained results have some practical implications. In the research and reality of the operation of present energy systems, meteorology starts to play a very important and, in many instances, crucial role in enabling and securing an efficient and reliable operation of the power system. The need for including meteorology-based studies in energy research comes directly from the unprecedented increase in the installed capacity of renewable energy sources, especially the ones in which variability/availability is driven by climate and weather [21]. The Polish power sector is starting a process of transformation. The increasing share of renewables such as wind and, in particular, solar energy (as observed in 2019/2020) is driven by an increasing awareness of the energy sector's impact on the natural environment, the growing cost of electricity generation from conventional fuels, the decreasing cost of renewables, and national/international policy. The power system expansion/development studies [22] call for data with both higher temporal and spatial resolution. This can be provided by either ground measurements, satellite measurements, numerical weather prediction models, or reanalysis data sets. In this paper we have investigated whether the in-house data available from a governmental institution (Institute of Meteorology and Water Management) can be used to simulate aggregated PV generation on a country level. The results

obtained proved the high value of the already available data and its promising applications in power system expansion studies as well as research dedicated, for example, to optimal location of PV systems from the perspective of grid topology or the impact of PV systems on the residual load curve.

This study intended to present the results of the first tests on the freshly published data by the Polish Transmission System Operator—PSE. Therefore, we did not go into detail comparing different statistical or machine learning techniques. Clearly, for now, the data sample is relatively short and, at the time of writing, was limited to one month. In the future, we plan to extend this research by investigating (a) in detail the impact of the input set on the model performance, (b) comparing different simulation tools, (c) the impact of a long series of meteorological parameters on the quality of the model, and finally (d) selecting a minimal set of representative meteorological stations sufficient for simulations. The results presented here should be taken with caution since solar irradiation has a high annual variability, and model performance might, therefore, vary depending on the part of the year.

**Author Contributions:** Conceptualization, J.J. and M.W.; methodology, J.J.; software, J.J.; validation, M.W., M.F.; resources, M.W.; data curation, M.W.; writing—original draft preparation, J.J.; writing—review and editing, M.W. and M.F.; visualization, J.J.; supervision, M.F.; project administration, M.F. All authors have read and agreed to the published version of the manuscript.

**Funding:** This research received no external funding.

**Conflicts of Interest:** The authors declare no conflict of interest.

### Nomenclature

| | |
|---|---|
| ANN | Artificial Neural Network |
| ECMWF | European Centre for Medium-Range Weather Forecasts |
| ERA5 | ECMWF Reanalysis 5th Generation |
| IMWM-NRI | Institute of Meteorology and Water Management—National Research Institute |
| MAPE | Mean Absolute Percentage Error |
| MERRA | Modern Era Retrospective-Analysis for Research and Applications |
| NN | Neural Network |
| PS | Power System |
| PSE | pol. Polskie Sieci Elektroenergetyczne—Polish Transmission System Operator |
| PV | Photovoltaic |
| RMSE | Root-Mean-Squared Error |
| VRES | Variable Renewable Energy Sources |

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
