# Peer review of "Simulating Power Generation from Photovoltaics in the Polish Power System Based on Ground Meteorological Measurements—First Tests Based on Transmission System Operator Data"

_energies, doi:10.3390/en13164255_

Round 1
Reviewer 1 Report
The acronyms (PSE, PS, MAPE, RMSE) should be explained.
L115: What reresents the figure 2? How was obtained? What software was used?
L136: How was obtained the figure 4? What type of regression is discussed? all, training, test or validation?
The authors should add more comments about figures 4 and 5.
Conclusions should be more comprehensive.
Author Response
Thank you for your comments regarding our manuscript. Please see attached file for detailed responses.

Reviewer 2 Report
An artificial neural network is used to solve the problem, but the method is not described in more detail. The significance of the article is mainly in the possibility of using the artificial neural network method also in other areas. I recommend publishing the paper, because similar articles concerning other territories (e.g. Mediterranean) have also been published.
I recommend add this work:
- Renno, F. Petito, A. Gatto, "ANN model for predicting the direct normal irradiance and the global radiation for a solar application to a residential building,"
Journal of Cleaner Production 135 (2016) 1298e1316
www.elsevier.com/locate/jclepro
Author Response

(The authors gave the same response as above.)

Reviewer 3 Report
In this paper, the authors introduced a new exploring how well those can be modelled based on the meteorological measurements provided by the institute of meteorology and water management. Hourly time series of PV generation on a country level and irradiation, wind speed and temperature measurements from meteorological stations covering one month are used as inputs to create an artificial neural network. The idea behind this is interesting. However, I still have quite a number of concerns in this manuscript. There are times where there are not enough data to support the conclusions of the author. Please see some of the major concerns below.
1.The information for the Observed irradiation in 23 stations along with PV production on a country level on 1st of May 2020 is not clear. The authors should give much more information about this. So the readers can get its reproducibility.
2.The authors should give much more information about the novelty of this paper, especially the effect of using this new artificial neural network, which applications can be used this network?
3.More references need to be included in the introduction part to understand the applications of using an optical neural network such as:
a."Color image identification and reconstruction using artificial neural networks on multi-mode fiber images: Towards an all- optical design"Optics Letters, 43(22), 2018 (5603-5606)
b."Neural networks within multi-core optic fibers",Scientific Reports 6, 2016 (article no.29080)
4. Much more discussion about the results should be given in this paper, especially the author needs to provide enough physicals mechanism analysis about the results.
Author Response

(The authors gave the same response as above.)

Round 2
Reviewer 1 Report
I recommend this paper for publication.
Reviewer 3 Report
The new version can be published.